# Beta Lactams Plus Daptomycin Combination Therapy for Infective Endocarditis: An Italian National Survey (BADAS)

**DOI:** 10.3390/antibiotics11010056

**Published:** 2022-01-02

**Authors:** Silvia Corcione, Tommaso Lupia, Carlo Pallotto, Daniele Roberto Giacobbe, Ilaria De Benedetto, Giacomo Stroffolini, Simone Mornese Pinna, Carlo Tascini, Matteo Bassetti, Francesco Giuseppe De Rosa

**Affiliations:** 1Department of Medical Sciences, University of Turin, 10126 Turin, Italy; silvia.corcione@unito.it (S.C.); ilaria.debenedetto@unito.it (I.D.B.); giacomo.stroffolini@unito.it (G.S.); simone.mornesepinna@unito.it (S.M.P.); francescogiuseppe.derosa@unito.it (F.G.D.R.); 2Department of Infectious Diseases, Tufts University School of Medicine, Boston, MA 02129, USA; 3Infectious Diseases Unit, Cardinal Massaia Hospital, 14100 Asti, Italy; 4Infectious Diseases Unit, University Hospital of Perugia, 06156 Perugia, Italy; pallottoc@gmail.com; 5Department of Health Sciences (DISSAL), University of Genoa, 16132 Genoa, Italy; danieleroberto.giacobbe@unige.it (D.R.G.); matteo.bassetti@unige.it (M.B.); 6Clinica Malattie Infettive, San Martino Policlinico Hospital-IRCCS for Oncology and Neurosciences, 16132 Genoa, Italy; 7Department of Medicine, University of Udine, 33100 Udine, Italy; c.tascini@gmail.com; 8Infectious Diseases Clinic, Udine University Hospital, 33100 Udine, Italy

**Keywords:** endocarditis, daptomycin, combination therapy, survey, gram-positive

## Abstract

Background: infective endocarditis (IE) remains a severe disease frequently encountered in clinical practice and often requiring interdisciplinary medical and surgical management. This national survey aims to describe the clinical prescribing habits of the use of daptomycin in the setting of IE and the possible role for combination therapy with beta-lactams. Methods: The study was a cross-sectional internet-based questionnaire survey on therapy with daptomycin. The questionnaire was designed with closed-ended questions and distributed using the SurveyMonkey^®^ platform between October 2019 to December 2020. Results: 55 clinicians from twelve Italians regions joined the questionnaire. The survey reported use of daptomycin as first-line choice in 31.48% of cases and as the first-line anti-MRSA agent in 44.44%. The empiric use of daptomycin was stated in the high suspicion of MRSA rather than MSSA, enterococcal or streptococcal IE. The rationale of daptomycin for the empirical treatment of native and prosthetic valve IE was mostly the possibility of administering an aminoglycoside-sparing combination regimen, high bacterial killing rate and high clinical efficacy. Conclusions: In conclusion, in selected patients, daptomycin could be a feasible option for the treatment of infective endocarditis in line with data from the European registry of daptomycin.

## 1. Introduction

Infective endocarditis (IE) remains a severe disease frequently encountered in clinical practice and often requiring interdisciplinary medical and surgical management [1,2,3,4,5].

Recently, the European Society of Cardiology (ESC) EURObservational Research Programme (EORP) implemented the European Endocarditis (EURO-ENDO) international registry, confirming the high prevalence of gram-positive microorganisms involved in both native and prosthetic valve endocarditis [6,7]: *Staphylococcus* spp., oral *streptococci, Enterococcus* spp., and *Streptococcus gallolyticus* are the most common pathogens in this registry [6,7].

Furthermore, the IE population has been transforming, with increasing incidence in elderly adults, highly comorbid patients and prosthetic heart valve cases [1,2,3,6,7]. IE populations often have multiple comorbidities, such as cardiovascular diseases, diabetes mellitus, cancer, and chronic liver and kidney diseases [1,2,3,6,7]. IE in this population requires effective and manageable therapies according to the frailty. Frail patients are characterized by high morbidity and mortality and puts the subjects at greater risk of developing adverse antibiotic effects [1,2,3,6,7].

Daptomycin, a lipopeptide antimicrobial with concentration-dependent bactericidal activity against gram-positive organisms, including antimicrobial-resistant pathogens such as methicillin-resistant *Staphylococcus aureus* (MRSA), is approved for treating complicated skin and soft tissue infections [cSSSI], furthermore clinical experiences and preliminary data are increasing and are favourable to treating osteoarticular infections [8]. Daptomycin was originally found non-inferior to standard therapy for *S. aureus* bacteremia and right-sided endocarditis at the daily dose of 6 mg/kg. [9]. Later on, Daptomycin was successfully used at median dose of 5.93 mg/kg for vancomycin-resistant *Enterococcus* spp. (VRE) bloodstream infections, and it has been recently used in the therapy of both left and right-sided IE [10,11,12]. Nevertheless, the use of standard dosing of 6 mg/kg once-daily has led to the emergence of resistant strains associated with clinical failures and, more recently, higher doses (8–12 mg/kg once daily) provided improved effectiveness in the treatment of Gram-positive endocarditis [10,11,12].

Considering daptomycin in IE treatment represents an option to encourage aminoglycoside and glycopeptide sparing strategy in order to avoid renal damage, especially in elderly comorbid population [12,13]. Moreover, daptomycin has been demonstrated safe and effective when vancomycin minimum inhibitory concentration (MIC) values rise above 1 mg/L obtaining faster clearance of bacteremia and lower rates of clinical failure [14] Observational and randomized controlled trials suggested the combination of daptomycin with a beta-lactam in patients with MRSA bacteremia [15,16,17,18] Nonetheless, a recent metanalysis including 1636 patients in this setting showing higher cure rates with the combination therapy failed to demonstrate a difference in mortality rates or renal damage because of the heterogeneity in the choice of the beta-lactam [19]. Finally, some uncertainty on the daptomycin combination therapy with beta-lactams remains in the contest of IE.

This national survey aims to describe the clinical prescribing habits of the use of daptomycin in the setting of IE and the possible role for combination therapy with beta-lactams.

## 2. Results

Fifty-five clinicians joined the questionnaire through electronic mail. The hospitals and clinicians involved are distributed in twelve Italian regions (Calabria Campania, Emilia-Romagna, Friuli Venezia Giulia, Lazio, Liguria, Lombardia, Liguria, Toscana, Piemonte, Sicilia, Umbria and Veneto). Of these, 21.85% (28) were university hospitals, 9.26% (5) were base, 29.63% (16) were first, and 9.26% (5) were second-level care hospitals. The main features of centers involved in the survey are described in Table 1. The number of beds in each center was, for the relative majority, higher than 500 (50.91%; 28), followed by 200–500 (36.36%; 20), 100–200 (7.27%; 4), and fewer than 100 beds (5.45%; 3). The majority of the hospitals (59.26%; 32) had high-risk units on-site with 18.52% (10) being cardiosurgical, 12.96% (7) solid organ transplant, 5.56% (3) (Table 1).

An in-hospital infectious disease (ID) consultant was available in (29.63% (16) of the hospitals, and a consultant and ID unit was present in 66.67% (36) (Table 2).

Almost all of the hospitals (98.15%; 53) had an infection committee with access to epidemiological MDRO data (96.3%; 53), and more than half of the participating centres reported the presence of antibiotic guidelines (64.81%; 35), both locally (71.43%; 25) and regionally (28.57%; 10) (Table 2).

73.58% of center reported an incidence of <50 cases of IE/year (Table 3). Cardiosurgical service was available in 59.26% (32) hospitals, whilst the endocarditis team was reported in 70% (37) of them (Table 3).

The ID consultant was responsible for prescribing empiric antibiotic IE therapy in 62.97% (34) of the hospitals (Table 3). The preferred empirical treatment was more often a combination of antibiotics than monotherapy in both native 77.77% (42) and prosthetic 83.33% (45) valve IE. In both IE subtypes, respondents reported regimens always including vancomycin (5.55%; 3 and 5.88%; 3 respectively) or aminoglycoside (1.88%; 1 and 3.92%; 2 respectively). The duration of treatment was ≤6 weeks and according to infection isolation rather than >6 weeks independent of isolation, either in case of need (43.33% (45) vs. 16.67% (9)) or no need (83.33% (45) vs. 16.67% (9)) of cardiosurgical treatment. The median duration of antibiotic treatment in native and prosthetic valve IE was 4 and 6 weeks, respectively (IQR 4–12) (Table 3).

The interviewed clinicians reported the use of daptomycin as first-line choice in 31.48% (17) of cases and as the first-line anti-MRSA agent in 44.44% (24). The empiric use of daptomycin was stated in the high suspicion of MRSA (90.38%; 47) rather than MSSA (3.85%; 2), enterococcal (5.77%; 3), or streptococcal (0%; 0) IE (Table 4).

The rationale in the use of daptomycin for the empirical treatment of native and prosthetic valve IE was: the possibility of administering an aminoglycoside-sparing combination regimen (24.53%; 13 and 9.80%; 5, respectively), high bacterial killing rate (24.53%; 13% and 3.92%; 2, respectively); high clinical efficacy (18.87%; 10 and 9.80%; 5, respectively); high vegetation penetration (11.32%; 6 and 7.84%; 4, respectively); biofilm activity (9.43%; 5 and 56.86%; 29, respectively); safety and low toxicity (5.66%; 3 and 9.80%; 5 respectively); and single-day OPAT administration (5.66%; 3 and 1.96%; 1, respectively). 

High-dose daptomycin (8–10 mg/kg) in combination treatment (70.37%; 38) was preferred. Combination regimens most often included beta-lactams (54.17%; 26), followed by novel cephalosporins, such as ceftaroline and ceftobiprole (18.75%; 9), rifampin (10.42%; 5), 3rd- or 4th-generation cephalosporins (8.33%; 4), or aminoglycosides (8.33%; 4). In targeted therapy, daptomycin was frequently used for MRSA IE (61.11%; 33), MRSA or MSSA IE (16.67%; 9), or independently against Streptococcus, MRSA, or MSSA IE (14.81%; 8). It was less commonly used in enterococcal IE (7.41%; 4).

More specifically, daptomycin was considered in combination with beta-lactams more often in MRSA IE (46.30%; 25), MRSA and MSSA IE (18.52%; 10), and independently against Streptococcus, MRSA, or MSSA IE (12.96%; 7). 

A slightly higher use of daptomycin plus beta-lactams over monotherapy was recorded for enterococcal IE (18.52%; 10), and this combination was used, especially as a targeted therapy during hospitalization followed by oral de-escalation (35.85%; 19) and OPAT with the withdrawal of beta-lactams after discharge (26.42%; 14).

In patients with stable disease or after surgical treatment, de-escalation, stop or switch to monotherapy was considered after 1–2 (40.0%; 22) or 3 weeks (32.73%; 18) of treatment. For 23% of the participating hospitals, de-escalation was not routinely performed. Possible de-escalation options were: beta-lactams, cephalosporins, TMP/SMX, clindamycin, doxycycline, rifampin, and fluoroquinolone for MSSA and TMP/SMX, dalbavancin, linezolid, doxycycline, rifampin for MRSA. (Table 4).

## 3. Discussion

Our survey aimed to explore the prescribing habits for the management of IE involving the use of daptomycin, with a particular interest in beta-lactam plus daptomycin combination therapy for IE, as prescribed by ID and internal medicine physicians in Italy between 2019 and 2020.

Respondents from different hospitals located in 13 regions covered homogeneous Italian territory (southern, central and northern Italy). Our results showed that respondents mainly worked within large (≥500 beds) university hospitals that frequently included an ICU and neuro/cardio-surgical unit, and they had access to ID consultants or antibiotic guidelines. The incidence of IE appeared to be low, and possibly, as a consequence, endocarditis teams were not widespread in our sample. Interestingly, cardiac-surgeons were largely available in hospitals from which we retrieved data, which may provide a window of opportunity to create endocarditis teams with a multidisciplinary and comprehensive approach, as suggested by current guidelines [1,2,3,4,5]. In fact, the benefit of an endocarditis team has been well established in reducing in-hospital and long-term mortality, improving antimicrobial therapy adherence and reducing deaths caused by embolic events and multiple organ failure. [1,2,3,4,5,20].

Empirical combination therapy was preferred over monotherapy as a starting regimen, regardless of native or prosthetic valve IE. Treatment duration was four to six weeks, with no difference regarding the surgical infection control culture-negative valve. This appears to be in line with IE guidelines [1,2]. Interestingly, almost all respondents considered vancomycin or an aminoglycoside unnecessary in the regimen, providing the opportunity for opening up the possibility of fewer nephrotoxic treatments that spare these molecules. Daptomycin was primarily preferred over other MRSA IE therapies (90%) and was the first line in the empiric approach in a third of cases. 

Combination therapy, including high-dose daptomycin, was preferred over monotherapy, mainly adding beta-lactams or novel cephalosporins. This data might be explained by the in vitro synergistic effect of these compounds and the emerging data on higher rate of clinical failure associated with the emergence of non-susceptible strains in the contest of infection with high inoculum treated with daptomycin monotherapy, such as MRSA persistent bacteremias [21,22,23]. In vitro studies reported that combination of daptomycin and β-lactams could change the superficial charge of cell envelope finally enhancing daptomycin-mediate killing and boosting the cationic antimicrobial peptides-mediate killing even against VRE strains [24,25].

The combination of daptomycin plus beta-lactam in MRSA BSI, especially when including anti-MRSA novel cephalosporins, achieved a clinical success rate ranging from 60 to 100% [15,16,18,19,26]. In another recent study, aminoglycosides were rarely prescribed with daptomycin, and AG-sparing combination regimens were stated as one of the main reasons for choosing daptomycin plus beta-lactams in native IE likely with regard to the synergistic effect in enterococcal etiology. This is noteworthy given the higher mortality rate registered in patients with bacteraemia or IE and renal failure [26,27].

Lower dosage daptomycin was generally less common, both for combination and monotherapy. This can be explained by the fact that higher doses result in achieving better microbiological outcomes. Because of this, many clinicians have stated that the choice of daptomycin is driven by the possibility of achieving good bactericidal, high clinical efficacy and anti-biofilm activity in prosthetic endocarditis.

According to respondents, in one-third of cases, a daptomycin plus beta-lactam regimen after hospitalisation allowed oral de-escalation or attendance at an outpatient antimicrobial therapy clinic (OPAT) with the withdrawal of beta-lactam within one to two weeks for patients with stable disease or after surgical treatment. 

In this survey, considered de-escalation options are heterogeneous, including oral [28,29] and long-acting injectable agents, such as dalbavancin, as successfully observed in recent studies [30,31]. Interestingly, for one out of five participants, de-escalation was not commonly performed, probably highlighting difficulties in the follow up of patients outside the hospital setting.

This survey has different limits: respondents have been selected on a voluntary basis from the group of the SITA and not directly from the hospital setting. A possible bias towards respondents who may have a fair knowledge on the subject matter and. By using closed-ended questions to assess the level of knowledge on daptomycin use and IE management, respondents may have selected the most favourable answer instead of using qualitative methods to revel misconceptions.

## 4. Materials and Methods

### 4.1. Survey Design

The study was a cross-sectional internet-based questionnaire survey on therapy with daptomycin. The questionnaire was designed with closed-ended questions and distributed using the SurveyMonkey^®^ platform (San Mateo, CA, United States). We requested information on the participant’s specialty, hospital name, and size and type of hospital. Furthermore, we requested information regarding infectious diseases (ID) service in their hospital, antibiotic guidelines and stewardship programmes. Questions on incidence and management of IE caused by different Gram-positive bacteria were submitted to clinicians with focus on monotherapy or combination therapy for IE. Finally, the use of daptomycin, alone or in combination was investigated (the full questionnaire is available in the Appendix A).

The questionnaire was developed by three primary investigators (TL, SC, CP, CT and FGDR) and pre-tested by all authors for clarity and technical functionality.

Our target population were ID and internal medicine physicians treating patients, giving advice on antibiotic treatment in Italy and members of the Italian Society of Antimicrobial Therapy “Società Italiana di Terapia Anti-infettiva” (SITA and SITA GIOVANI).

We asked respondents to reply by describing the most common actual practice at their hospital. Only one participant from each hospital was included.

### 4.2. Survey Administration

One investigator submitted the proposal to join the questionnaire through electronic mail. After a positive response, an invitation was sent by the survey coordinator.

Participants were able to access the questionnaire multiple times to allow possible changes and completion later. The survey was voluntary, with no incentives offered to participants (other than being listed as a collaborator).

Participants have had three months (from October 2019 to December 2020) to access the questionnaire.

The survey and the questions did not require a mandatory answer but could also be skipped.

### 4.3. Response Rates

Response rates were calculated as numbers of clinicians from which an answer was recorded. Information on hospital names was used to screen for duplicate entries, but all data were subsequently anonymized for the analyses.

### 4.4. Statistical Analysis

Both completed and partially completed questionnaires were analysed using the number of completed responses per item as the denominator.

## 5. Conclusions

In conclusion, daptomycin could be a feasible option for the treatment of high inoculum intravascular infections (i.e., bacteremia with metastatic complications, endocarditis), especially in selected patients who had a higher risk of MRSA invasive infections, carrying prosthetic valves which required a drug with high bactericidal effect on biofilm [32]. High doses and combination therapy with beta-lactams, including ceftaroline or ceftobiprole, are more likely to be chosen. Moreover, a prospective interventional study comparing daptomycin plus beta-lactams should be considered to evaluate outcomes.

## Figures and Tables

**Table 1 antibiotics-11-00056-t001:** Features of the centers enrolled in the survey.

Type of Hospital	University	Base Level	I Level	II Level
(Responding 54/55 partecipants)	28 (51.85%)	5 (9.26)	16 (29.63)	5 (9.26)
Number of beds	<100	100–200	200–500	>500
(Responding 55/55 partecipants)	3 (5.45%)	4 (7.27)	20 (36.36)	28 (50.91)
Intensive Care Unit in-hospital	No	Yes		
(Responding 55/55 partecipants)	2 (3.64%)	53 (96.36)		
		**If YES category:**		
		*CCH/cardio*; 16 (30.18)		
		*NCH/neuro*; 6 (11.32)		
		*Burn*; 1 (1.88)		
		*Transplant*; 5 (9.43)		
		*Pediatric*; 3 (5.66)		
High Risk Unit in-hospital	No	Yes		
(Responding 54/55 partecipants)	32 (59.26%)	22 (40.74)		
		**If YES category:**		
		*Haematologic*; 3 (5.56)		
		*SOT*; 7 (12.96)		
		*Pediatric*; 1 (1.85)		
		*NCH*; 1 (1.85)		
		*CCH*; 10 (18.52)		

CCH: cardiac surgery; NCH: neurosurgery; SOT: solid organ transplant.

**Table 2 antibiotics-11-00056-t002:** Information regarding infectious disease specialist and infection control procedures.

ID Consultant	In-Hospital	In-Hospital with ID Unit	On-Call	No ID
(Responding 54/55 partecipants)	16 (29.63%)	36 (66.67)	1 (1.85)	1 (1.85)
Infection Comitee in the Hospital	No	Yes		
(Responding 54/55 partecipants)	1 (1.85%)	53 (98.15)		
Alert systems for MDROs	No	Yes		
(Responding 54/55 partecipants)	42 (77.78%)	12 (22.22)		
Available epidemiological MDRO’s data	No	Yes		
(Responding 54/55 partecipants)	2 (3.70%)	54 (96.3)		
Presence of antibiotic GL	No	Yes	Regional GL	Hospital GL
(Responding 54/55 partecipants)	19 (35.19%)	35 (64.81)	10 (28.57)	25 (71.43)

ID: Infectious Diseases; MDROs: multi-drug resistant organisms; GL: guidelines.

**Table 3 antibiotics-11-00056-t003:** Incidence on endocarditis management from enrolled hospitals.

**IE Per Years**	**>50 Years**	**<50 Years**	
(Responding 53/55 partecipants)	14 (26.42%)	39 (73.58)	
**Presence of Endocarditis Team**	**No**	**Yes**	
(Responding 53/55 partecipants)	37 (69.81%)	16 (30.19)	
**Presence of CCH**	**No**	**Yes**	
(Responding 54/55 partecipants)	22 (40.74%)	32 (59.26)	
**Prescription of ABX in IE**	**ID**	**Physician**	**Both**
(Responding 53/55 partecipants)	34 (62.97%)	1 (1.85)	18 (33.33)
**Empiric ABX in IE Native Valve**	**Combination**	**Monotherapy**	
*Total*	42 (77.77%)	12 (22.22)	
*Including always Vancomycin*	3 (5.55%)		
*Including always AG*	1 (1.88%)		
**Empiric ABX in IE Protesic Valve**	**Combination**	**Monotherapy**	
*Total*	45 (83.33%)	1 (1.85)	
*Including always Vancomycin*	3 (5.88%)		
*Including always AG*	2 (3.92%)		
**Duration of treatment (No need for CCH)**	**<6 wks (according to isolated)**	**>6 wks (indipendent to isolated)**	
(Responding 54/55 partecipants)	45 (83.33%)	9 (16.67)	
**Duration of treatment (Need for CCH)**	**<6 wks (according to isolated)**	**>6 wks (indipendent to isolated)**	
(Responding 54/55 partecipants)	45 (83.33%)	9 (16.67)	
**Duration of treatment**	**IE Native Valve**	**IE Protesic Valve**	
*Median (IQR)*	4 (2–89) wks	6 (4–12)	

IE: infective endocarditis; ABX: antibiotics; CCH: cardiac surgery; AG: aminoglycoside.

**Table 4 antibiotics-11-00056-t004:** Empirical, targeted combination therapy or monotherapy including Daptomycin in different settings: responses from enrolled hospitals.

**Empiric Daptomycin in IE**	**IE MRSA**	**IE MSSA**	**IE *Enterococcus***	**IE *Streptococcus***
(Responding 52/55 partecipants)	47 (90.38)	2 (3.85)	3 (5.77)	0 (0)
**Ongoing Resistant *Streptococcus* spp. strains to Daptomycin**	**No**	**Yes**		
(Responding 52/55 partecipants)	40 (76.92)	12 (23.07)		
Monotherapy		11 (91.66)		
Combination		1 (8.34)		
**Daptomycin in Clinical Practice in IE**	**First-line in Empiric Therapy**	**First-line in MRSA IE**	**No First-line**	
(Responding 53/55 partecipants)	17 (31.48)	24 (44.44)	12 (24.07)	
**Daptomycin Empirical in IE (Native Valve)**				
High Clinical Efficacy	10 (18.87)			
Combination with AG-sparing	13 (24.53)			
High penetration in vegetations	6 (11.32)			
Safety and low toxicity	3 (5.66)			
Single-day administration and OPAT choice	3 (5.66)			
Biofilm acitivity	5 (9.43)			
High bacterial killing rate	13 (24.53)			
Low rate of resistant strains/favourable MICs	0 (0)			
**Daptomycin Empirical in IE (Prostethic Valve)**				
High Clinical Efficacy	5 (9.80)			
Combination with AG-sparing	5 (9.80)			
High penetration in vegetations	4 (7.84)			
Safety and low toxicity	5 (9.80)			
Single-day administration and OPAT choice	1 (1.96)			
Biofilm acitivity	29 (56.86)			
High bacterial killing rate	2 (3.92)			
Low rate of resistant strains/favourable MICs	0 (0)			
**Methods of Daptomycin Using in IE**				
(≤6 mg/kg) Monotherapy	0 (0)			
(≤6 mg/kg) Combination	4 (7.41)			
(8-10 mg/kg) Monotherapy	12 (22.22)			
(8-10 mg/kg) Combination	38 (70.37)			
**Daptomycin in combination therapy in IE**				
Rifampin	5 (10.42)			
AG	4 (8.33)			
Beta-lactams	26 (54.17)			
Cephalosporins (III or IV gen)	4 (8.33)			
Novel Cephalosporins	9 (18.75)			
**Daptomycin dose in clinical practice in IE according to pathogen**				
*Streptococcus* spp.	0 (0)			
MSSA	0 (0)			
MRSA	33 (61.11)			
Both MRSA and MSSA	9 (16.67)			
*Enterococcus* spp.	4 (7.41)			
Indipendently to pathogens	8 (14.81)			
**Daptomycin plus Beta-lactams in IE**				
*Streptococcus* spp.	0 (0)			
MSSA	2 (3.70)			
MRSA	25 (46.30)			
Both MRSA and MSSA	10 (18.52)			
*Enterococcus* spp.	10 (18.52)			
Indipendently to pathogens	7 (12.96)			
**Daptomycin plus Beta-lactams in IE (regimens)**				
Empiric therapy	9 (16.98)			
Targeted plus OPAT	14 (26.42)			
Targeted then oral de-escalation	19 (35.85)			
Targeted in-hospital (no OPAT)	11 (20.75)			
**De-escalation after Daptomycin therapy in IE (stable/operated)**				
After 1–2 wks	22 (40.00)			
After 3 wks	18 (32.73)			
After 6 wks	3 (5.45)			
De-escalation is not common in our centre	12 (21.82)			
**Choices for De-escalation after Daptomycin in IE**				
MSSA	Beta-lactams, cephalosporins, TMP/SMX, clyndamicin, doxicycline, doxicycline, rifampicin and fluoroquinolones			
MRSA	TMP/SMX, dalbavancin, linezolid, doxicycline and rifampicin			
**Daptomycin interruption or substitution**				
Partial or no response	1 (1.85)			
Adverse effects	7 (12.96)			
Costs	3 (5.56)			
Medications more accessible in OPAT or long-term facility	23 (42.59)			
De-escalation	20 (37.04)			

IE: infective endocarditis; MRSA: methicillin-resistant *Staphylococcus aureus*; MSSA: methicillin-susceptible *Staphylococcus aureus*; OPAT: outpatient therapy; TMP/SMX: trimethoprim/sulfametoxazole; AG: aminoglycoside; MIC: minimum inhibitory concentration.

## Data Availability

The data presented in this study are available on request from the corresponding author.

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
