# Peer review of "Beta Lactams Plus Daptomycin Combination Therapy for Infective Endocarditis: An Italian National Survey (BADAS)"

_antibiotics, 2022, doi:10.3390/antibiotics11010056_

Round 1

Reviewer 1 Report

The investigators describe a survey of Italian care-givers on the topic of antimicrobial therapy for infective endocarditis. The design appears to be adequate to the task. The results are presented clearly. The authors do not editorialize; rather, they present the results and summarize the observations. 

Author Response

Dear Editor,

Thanks for the opportunity to revise our manuscript. Please find the response to reviewers’ comments on our manuscript entitled “Beta lactams plus Daptomycin combination therapy for infective endocarditis: An Italian National Survey (BADAS)to be considered for publication in Antibiotics Journal

Reviewer comments

Reviewer  1#

The investigators describe a survey of Italian care-givers on the topic of antimicrobial therapy for infective endocarditis. The design appears to be adequate to the task. The results are presented clearly. The authors do not editorialize; rather, they present the results and summarize the observations.

Dear reviewer, thank you for these comments and your revision

Reviewer 2 Report

The manuscript was well written. In order to really detect the importance of the use of daptomycin in infective endocarditis, the study design must be improved, looking at outcome and comparing different antibiotic strategies. 

Author Response

Dear Editor,

Thanks for the opportunity to revise our manuscript. Please find the response to reviewers’ comments on our manuscript entitled “Beta lactams plus Daptomycin combination therapy for infective endocarditis: An Italian National Survey (BADAS)to be considered for publication in Antibiotics Journal

Reviewer comments

Reviewer  2#

The manuscript was well written. In order to really detect the importance of the use of daptomycin in infective endocarditis, the study design must be improved, looking at outcome and comparing different antibiotic strategies.

Dear Reviewer, thank you for this comment. We appreciate the fact that the paper is considered well written. From a methodological point of view, this is a nationwide cross-sectional internet-based questionnaire survey on antimicrobial regimens including daptomycin. The questionnaire was designed with closed-ended questions and distributed using the SurveyMonkey® platform. The aim of the study was to assess the current knowledge, real-life perception of and approach to daptomycin combination or monotherapy.  We requested information on the participant's speciality, hospital name, and size and type of hospital. Furthermore, we requested information regarding Infectious diseases (ID) service in their hospital, antibiotic guidelines and stewardship programmes. Our target population were ID and internal medicine physicians treating patients, giving advice on antibiotic treatment in Italy and members of the Italian Society of Antimicrobial Therapy “Società Italiana di Terapia Anti-infettiva” (SITA and SITA GIOVANI).

Questions on incidence and management of IE caused by different Gram-positive bacteria were submitted to clinicians with a focus on monotherapy or combination therapy for infectious endocarditis (IE). Finally, the use of daptomycin, alone or in combination was investigated. According to that, it is not possible to detect any difference in outcomes nor further differences in therapeutic strategies, due to the nature of the study and the target of the survey. This survey is designed to gain knowledge on the current status of daptomycin’s prescriptive strategies and is not directed to answer the question on outcomes, but rather to give a picture frame of the tendency of the use of that molecule in Italian specialists’ groups. This is of high importance in view of programming upcoming multicenter works possibly designed with the aim of assessing outcomes and antibiotic strategies in real-life settings (prospective studies or comparatives) but cannot answer that question in and for itself.

Reviewer 3 Report

The time frame when the study was conducted should be mentioned in abstract or title

Tables need to be rearranged - the categories under the answer “yes”, for example, should be listed in the same collumn, and not separately. It is impossible to follow the results in the current form.

What was the response rate in the end? This information needs to be included.

Was the sample size calculated? If not, why? what is the power of the study? How many participants were eligable for the study?

Was the survey pretested on a small number of respondents or only in survey creators?

Author Response

Dear Editor,

Thanks for the opportunity to revise our manuscript. Please find the response to reviewers’ comments on our manuscript entitled “Beta lactams plus Daptomycin combination therapy for infective endocarditis: An Italian National Survey (BADAS)to be considered for publication in Antibiotics Journal

Reviewer comments

Reviewer  3#

  1. The time frame when the study was conducted should be mentioned in abstract or title

Dear reviewer thank you for this comment. We have added the study period in the Abstract section of the manuscript. You can find changes highlighted in the text.

  1. Tables need to be rearranged - the categories under the answer “yes”, for example, should be listed in the same collumn, and not separately. It is impossible to follow the results in the current form.

Dear reviewer, we have revised all the tables according to your suggestions

  1. What was the response rate in the end? This information needs to be included.

Dear reviewer thank you for this comment. We have added response rates in all the tables for questions where it was available

  1. Was the sample size calculated? If not, why? what is the power of the study? How many participants were eligable for the study?

Dear reviewer, thank you for this comment. From a methodological point of view, this is a nationwide cross-sectional internet-based questionnaire survey on therapy with daptomycin. The questionnaire was designed with closed-ended questions and distributed using the SurveyMonkey® platform. Acceptance of the questionnaire was on a voluntary basis. This survey is designed to gain knowledge on the current status of daptomycin’s prescriptive strategies and is not directed to answer the question on outcomes. For these reasons and for the methodology of the study a sample and the power of the study was not assessed

Was the survey pretested on a small number of respondents or only in survey creators?

Dear reviewer, thank you for this comment. The survey was pretested only within survey creators.

Round 2

Reviewer 2 Report

The manuscript is well written and investigates the role of usage of daptomycin in infective endocarditis in Italy.

Lines 32-34: How do you support the role that daptomycin is a safe and well tolerated option, if you have not looked into outcomes? Please rephrase this sentence.  

Author Response

Dear Editor,

Thanks for the opportunity to revise our manuscript. Please find the response to reviewers’ comments on our manuscript entitled “Beta lactams plus Daptomycin combination therapy for infec-tive endocarditis: An Italian National Survey (BADAS)to be considered for publication in Antibiotics Journal

Reviewer comments

Reviewer  1#

  1. The manuscript is well written and investigates the role of usage of daptomycin in infective endocarditis in Italy.

Dear reviewer, thank you for this comment. We are very glad it was appreciated

  1. Lines 32-34: How do you support the role that daptomycin is a safe and well tolerated option, if you have not looked into outcomes? Please rephrase this sentence. 

Dear reviewer, thank you for this comment. We agree with and we have rephrased the sentence accordingly to your suggestions

Reviewer 3 Report

Authors have adressed all of mine concerns

Author Response

Dear Editor,

Thanks for the opportunity to revise our manuscript. Please find the response to reviewers’ comments on our manuscript entitled “Beta lactams plus Daptomycin combination therapy for infective endocarditis: An Italian National Survey (BADAS)to be considered for publication in Antibiotics Journal

Reviewer comments

Reviewer  3#

  1. Authors have adressed all of mine concerns.

Dear reviewer, thank you for this comment. We are very glad it was appreciated